# Association of Uric Acid Albumin Ratio with Recurrence of Atrial Fibrillation after Cryoballoon Catheter Ablation

**DOI:** 10.3390/medicina58121872

**Published:** 2022-12-19

**Authors:** Mehmet Baran Karataş, Gündüz Durmuş, Ahmet Zengin, Murat Gökalp, Mert İlker Hayıroğlu, Tufan Çınar, Kadir Gürkan, Neşe Çam

**Affiliations:** 1Department of Cardiology, Health Sciences University, Siyami Ersek Cardiovascular and Thoracic Surgery Center, 34668 Istanbul, Turkey; 2Department of Cardiology, Health Sciences University, Sultan Abdulhamid Han Training and Research Hospital, 34668 Istanbul, Turkey

**Keywords:** catheter ablation, atrial fibrillation, uric acid/albumin ratio, recurrence

## Abstract

*Objective*: Despite improvements in the technology of catheter ablation of atrial fibrillation (AF), recurrences are still a major problem, even after a successful procedure. The uric acid/albumin ratio (UAR), which is an inexpensive and simple laboratory parameter, has recently been introduced in the literature as a predictor of adverse cardiovascular events. Hence, we aimed to investigate the relationship between the UAR and AF recurrence after catheter ablation. *Methods*: A total of 170 patients who underwent successful catheter ablation for AF were included. The primary outcome was the late recurrence after treatment. The recurrence (+) and recurrence (−) groups were compared for clinical, laboratory and procedural characteristics as well as the predictors of recurrence assessed by regression analysis. *Results*: In our study population, 53 (26%) patients developed AF recurrence after catheter ablation. Mean UAR was higher in the recurrence (+) group compared to recurrence (−) group (2.4 ± 0.9 vs. 1.8 ± 0.7, *p* < 0.01). In multivariable regression analysis, left atrial diameter (HR: 1.08, 95% CI: 1.01–1.16, *p* = 0.01) and UAR (HR:1.36, 95% CI: 1.06–1.75, *p* = 0.01) were found to be independent predictors of recurrence. In ROC analysis, the UAR > 1.67 predicted recurrence with a sensitivity of 77% and a specificity of 57% (AUC 0.68, *p* < 0.01). *Conclusion*: For the first time in the literature, the UAR were found to be correlated independently with AF recurrence after catheter ablation.

## 1. Introduction

The catheter ablation (CA) of atrial fibrillation (AF) has gained popularity over the last two decades, and recent scientific evidence recommended this strategy as a first line treatment option in selected individuals [1,2]. The technique has focused on isolating the triggering lesions, mainly located around the pulmonary veins ostia, by radiofrequency or cryo-balloon catheters. Despite improvements in this technology and the better understanding of pathophysiological mechanisms, freedom from AF after a successful ablation procedure can still be as low as 57% [3]. The recurrence rate after the CA of AF is estimated to be between 25–45% [4], and this is emerging as the primary drawback of this procedure. Recently, changes in the atrial architecture of atrial cardiomyopathy have been established to be related with inflammation [5]. Fibrosis, induced by inflammation, leads to conduction disturbances initiating and perpetuating the arrhythmia. Various inflammatory markers such as C-reactive protein (CRP), interleukin-2, interleukin-6, interleukin-8, and tumor necrosis factor α have shown to be associated with AF occurrence [6]. Uric acid (UA), an end product of purine metabolism, acts as a pro-oxidant molecule in high concentrations [7]. Many trials have clearly demonstrated the linkage between hyperuricemia and cardiac diseases or risk factors [8]. In addition, clinical evidence indicates that AF can occur due to apoptosis, cardiomyocyte hypertrophy and oxidative stress, which have been promoted by increased levels of UA [8]. In addition, a novel inflammatory marker, the uric acid/albumin ratio (UAR), showed a better predictive performance for AF development compared to its components [9]. Catheter ablation appears to provide better quality of life, symptom control and more sustainable sinus rhythm compared with antiarrhythmic drug therapy [10]. Hence, more reliable data are needed to identify patients who would benefit from this invasive procedure. In this study, our goal was to evaluate the role of UAR for the prediction of AF recurrence after CA.

## 2. Methods

### 2.1. Study Design and Patient Population

Patients with a history of drug refractory symptoms for paroxysmal or persistent AF, except those with tachycardia induced heart failure, and undergoing first time CA procedure between the years 2019–2021 were enrolled for this study. Patients with atrial flutter, atrial tachycardia or requiring any additional ablation procedures were not included. Patients with moderate and severe valvular heart disease and hypertrophic cardiomyopathy were also excluded given the high recurrence risk associated with these diseases. Sepsis, malignancy, or any condition associated with inflammation or malnutrition were also defined as exclusion criteria. The study protocol was approved by the relevant institutional review board.

### 2.2. Definitions and Study Outcome

Paroxysmal AF was identified as AF that terminates spontaneously or with intervention within seven days of onset. Persistent AF was regarded as episodes which lasted more than 7 days but less than a year and required termination by electrical or pharmaceutical cardioversion. The endpoint of the study was the late recurrence of AF after successful CA, which developed after a months blanking period. 24-h rhythm monitoring was routinely performed to all patients in control visits after the procedure. Symptomatic attacks requiring intervention or asymptomatic episodes of AF recorded on 24-h rhythm monitoring were defined as recurrences. Demographic characteristics, laboratory parameters, procedural features, complications, in-hospital mortality and long term follow-up data were collected from medical records.

Transoesophageal echocardiography (TEE) was conducted on all patients 6–12 h preceding CA to evaluate LA diameter and screen intracardiac thrombi. The ablation method was carried out while patients were anesthetized but without intubation. A transoesophageal echocardiography was conducted to perform trans-septal punctures under fluoroscopic guidance to determine the best position for puncture. Intravenous heparin was given following the trans-septal puncture to maintain an active clotting time of 300 to 400 s. A specialized mapping catheter was utilized to deliver a single transseptal puncture (Achieve^TM^, Medtronic, Minneapolis, MN, USA). A guidewire and a 12-Fr steerable sheath were applied to position the 28-mm cryo-balloon catheter (Arctic Front Advance^TM^, Medtronic, Minneapolis, MN, USA) (Flexcath, Medtronic Minneapolis, MN, USA). A 6-F decapolar coronary sinus catheter or a quadripolar diagnostic catheter was positioned in the superior vena cava for phrenic nerve stimulation while cryo-energy was delivered to the right pulmonary veins. The 240-s freezing cycle began after contrast injection revealed a pulmonary vein obstruction. Pulmonary vein isolation was assessed using a circumferential mapping catheter after two freezing cycles. The presence of both an entry and an exit block established pulmonary vein isolation. Transthoracic echocardiography was performed to all patients following pulmonary vein isolation to check pericardial effusion. Regardless of procedural effectiveness, anticoagulation was maintained for at least 3 months and was then dependent on the individual CHA_2_DS_2_-VASC score. Anticoagulation was discontinued in individuals with CHA_2_DS_2_-VASC scores of 0 in men and 1 in women. In the first 3-month period after ablation, the individuals continued on antiarrhythmic drugs they had been given prior to the procedure. Antiarrhythmic drugs were stopped 3 months after the procedure. The type of the antiarrhythmic drugs used was left to the discretion of the primary physician.

### 2.3. Statistical Analyses

All data were presented as a mean ± SD for variables with normal distribution or a median [inter-quantile range] for variables with non-normal distribution. Categorical variables were reported as numbers and percentages. Continuous variables were checked for the normal distribution assumption using Kolmogorov-Smirnov analysis. Categorical variables were tested by Pearson’s χ^2^ test and Fisher’s Exact Test. Differences between groups were evaluated using the Mann–Whitney U test or the Student’s t-test when appropriate. Univariable and multivariable binary logistic regression analyses were performed to investigate the independent correlates of recurrence of atrial fibrillation. As a result of the univariable regression analyses, variables which have *p* values < 0.10 were included in the multivariable regression analyses. Receiver operating curves were generated to define AUC and cut-off values for UAR. *p*-values were two sided and values <0.05 were considered statistically significant. All statistical studies were carried out using the Statistical Package for Social Sciences software (SPSS 22.0 for Windows, SPSS Inc., Chicago, IL, USA).

## 3. Results

The study population consisted of 170 subjects (mean age: 59.1 ± 11.7) who had undergone CA for AF. The median follow-up duration was 22 months (min: 7 months and max: 96 months) after procedure. The median time for the duration of AF was 17 months in the whole group. The type AF was mostly paroxysmal (71%). The mean CHA_2_DS_2_-VASC score of the patients was 1.8 ± 0.9. The mean procedural time was 123.9 min. The whole population was divided into two groups, namely recurrence and no recurrence. The frequency of the recurrence after ablation was 26% (n = 53). Groups compared for demographic and clinical features and these data were depicted in Table 1. Patients in the recurrence (+) group were older. The two groups were comparable in terms of chronic renal disease, chronic lung disease, smoking status, level of hemoglobin, creatinine, low density lipoprotein, thyroid stimulating hormone, and CRP. The procedural time did not differ between groups. Hypertension, coronary artery disease (CAD), congestive heart failure (CHF), and diabetes mellitus (DM) were common in the recurrence (+) group. The mean CHA_2_DS_2_-VASC score was higher in the recurrence (+) group compared to the recurrence (−) group (2.9 ± 1 vs. 1.4 ± 0.9, *p* < 0.01). Ninety percent of the study population were on a beta blocker or calcium channel blocker medication after the procedure. There was no difference between groups in terms of rate control and antiarrhythmic medications that commenced in the peri-procedural period. While AF duration was longer in the recurrence (+) group, the frequency of the paroxysmal AF was higher in the recurrence (−) group. When we look at the echocardiographic features, the frequency of the left ventricular hypertrophy (LVH) was similar between groups, but the mean left ventricular ejection fraction (LVEF) was lower and the mean left atrial (LA) diameter was higher in the recurrence (+) group. The mean UA levels were higher and the albumin levels were lower in the recurrence (+) group (8.1 ± 2.6 vs. 6.5 ± 2.1, *p* < 0.01; 3.5 ± 0.6 vs. 3.7 ± 0.5, *p* = 0.03 respectively). According to the Pearson correlation analyses, uric acid and albumin levels were negatively and significantly correlated (−0.69, *p* < 0.01). The mean UAR was higher in the recurrence (+) group compared to the other group (2.4 ± 0.9 vs. 1.8 ± 0.7, *p* < 0.01).

We performed univariable and multivariable binary logistic regression analyses for all variables in order to identify the independent predictors of recurrence after the procedure. In univariable regression analyses, LA diameter, UAR, CHA_2_DS_2_-VASC, AF type and AF duration were found to be correlated with recurrence (Table 2). When we entered these variables into the multivariable regression analysis, only LA diameter (HR: 1.08, 95% CI: 1.01–1.16 *p* = 0.01) and the UAR (HR:1.36, 95% CI: 1.06–1.75, *p* = 0.01) were ascertained as independent predictors of recurrence (Figure 1). We had also examined the Tolerance and Variance Inflation Factor (VIF) as a multicollinearity statistic, and we determined that the tolerance values were >0.1 and the VIF values were <10 for all parameters. Accordingly, there was no multicollinearity between each of the variables in the regression model.

In ROC analysis, the UAR > 1.67 predicted recurrence with a sensitivity of 77% and a specificity of 57% (AUC 0.68, *p* < 0.01), as shown in Figure 2. Patients with UAR > 1.67 had a 2.70-fold increased risk for recurrence compared to patients with UAR < 1.67 (HR:2.70, 95% CI: 1.41–5.15, *p* < 0.01). On a Kaplan-Meier curve, patients with UAR > 1.67 had significantly higher risk for recurrence (the log-rank, *p* < 0.01) (Figure 3). We also plotted the relationship between the probability of recurrence and UAR. The graphic demonstrates that a higher UAR indicates a higher probability of recurrence (Figure 4).

## 4. Discussion

To the best of our knowledge, the present study is the first to assess the role of the UAR for predicting the recurrence of AF after CA. The main finding of our study is that the UAR and LA diameter were independent predictors of AF recurrence following CA.

AF is the most common arrhythmia in the world, and multiple pathways contribute to it. The role of the pulmonary veins for triggers of AF is acknowledged. A previous model described multiple reentrant wavelets within the atrial tissue that contributed to the AF [11]. In addition, increased automaticity in the LA is believed to be responsible for the mechanism of AF [12]. The CA of AF is a well-established treatment modality for the rhythm control of AF. Studies have shown that the CA of AF is a safe and superior alternative to antiarrhythmic drugs for continued sinus rhythm and symptom improvement [13]. The CA is the best option for LVEF recovery and quality of life [14]. In addition, the association of CA with the decreased risk of stroke and mortality was shown in predominantly high risk patients in the literature [14].

Despite all of the advances, AF recurrence is considerably high in patients undergoing CA. The recurrence rate after CA of AF is estimated to be between 25–45% [4]. In our study, the recurrence of AF was 26%, which is in line with the literature. Many studies have been conducted to predict AF recurrence. The type of AF, advanced age, LA dimensions, smoking and female sex are predictors that increase AF recurrence [15,16]. We found LA diameter, AF duration, and AF type to be correlated with AF recurrence in univariable analysis. However, the correlation of duration and type of AF with recurrence were diminished after multivariable regression analyses. Furthermore, in our study the recurrence (+) group was older than the recurrence (−) group; however, regression analyses did not show any relationship between recurrence and age.

The previous research attributed low albumin levels to the development of CAD and the risk of mortality from any cause [17]. There is further evidence that lower serum albumin values may be connected to the recurrence of AF. On the other hand, UA has been the subject of many studies as an inflammatory marker [18]. Various studies determined the correlation between high levels of UA and cardiovascular risk in the general population [18]. Increased serum UA levels were reported to be independent predictors of the composite endpoints including cardiovascular death, myocardial infarction, and stroke in the entire population [19,20]. In addition, increased serum UA levels were found to be related with the development of AF. However, the exact mechanism has not been fully understood [21,22]. Oxidative stress, neurohormonal and inflammatory activation may all be responsible for AF occurrence. Oxidative stress may promote atrial electrical remodeling and the reentry mechanism of AF. The decreased level of nitric oxide and the shortening of the duration of action potential plateau phase and increasing repolarization velocity are potential mechanisms that may elucidate the relationship between uric acid and AF [23,24,25].

The UAR, which is an inexpensive and simple parameter that can be derived from laboratory findings, has recently been introduced in the literature as a predictor of adverse cardiovascular events. Çınar et al. established UAR as an independent predictor of the no-reflow phenomenon in patients with ST elevation myocardial infarction [26]. In another study, Selcuk et al. demonstrated UAR to be an independent predictor of new onset AF in patients with ST-elevation myocardial infarction [9]. They found 1.39 to be the cut-off value of UAR for predicting AF. Our cut-off UAR value for predicting AF is 1.67, close to that found in a previous study. While a previous study indicated 0.75 as the AUC value for UAR, our ROC analyses demonstrated an AUC value of 0.68; both of them show good discrepancy of UAR for predicting AF.

Our investigation has a few limitations. First of all, the study’s retrospective methodology limits its capacity to establish exact causality between UAR and the recurrence of AF. Secondly, although multivariable analyses were carried out, residual covariates may still be present. Thirdly, we did not collect the data of the baseline body mass index of all patients. Fourthly, because some recurrences were asymptomatic and detected in the 24-h rhythm monitoring, we could not differentiate whether they were paroxysmal or persistent in nature. Finally, more prospective studies should be conducted in the future to confirm the findings of our analysis.

## 5. Conclusion

In conclusion, we found that a higher UAR was associated with the increased frequency of AF recurrence after CA. Although CA is a safe and effective treatment strategy for AF, identifying patients at high risk for developing recurrence is of utmost importance. In this manner, as an easily calculable laboratory index, UAR may be utilized to predict the recurrence of AF after CA. Nevertheless, more studies are needed to confirm these findings and to determine whether UAR is of value for the therapeutic implication in patients with AF.

## Figures and Tables

**Figure 1 medicina-58-01872-f001:**
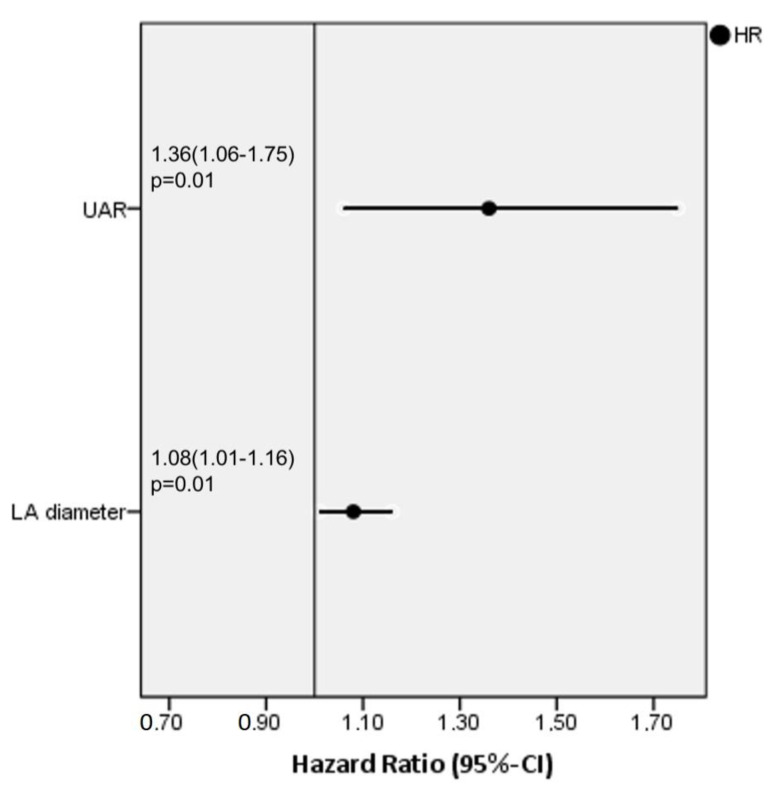
Forest plot graphic of independent predictors of atrial fibrillation recurrence after ablation according to the multivariable regression analyses.

**Figure 2 medicina-58-01872-f002:**
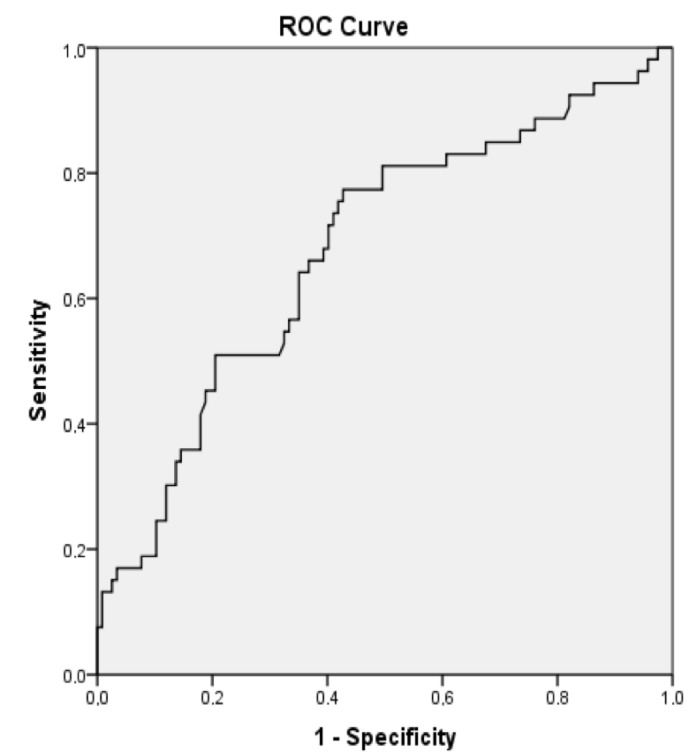
A receiver operating characteristic curve analyses of uric acid/albumin ratio (UAR).

**Figure 3 medicina-58-01872-f003:**
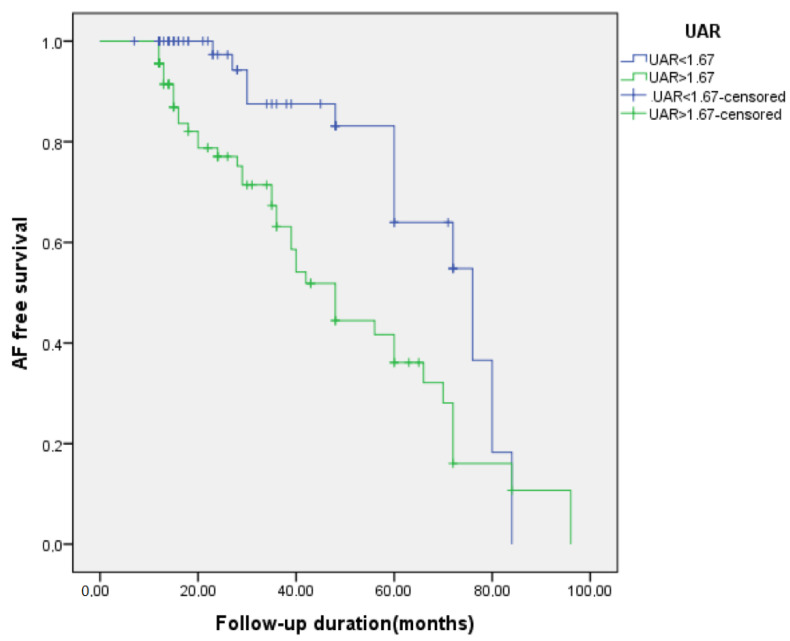
Kaplan-Meier curves comparing the AF free survival rate between patients with uric acid/albumin ratio (UAR) < 1.67 and UAR > 1.67.

**Figure 4 medicina-58-01872-f004:**
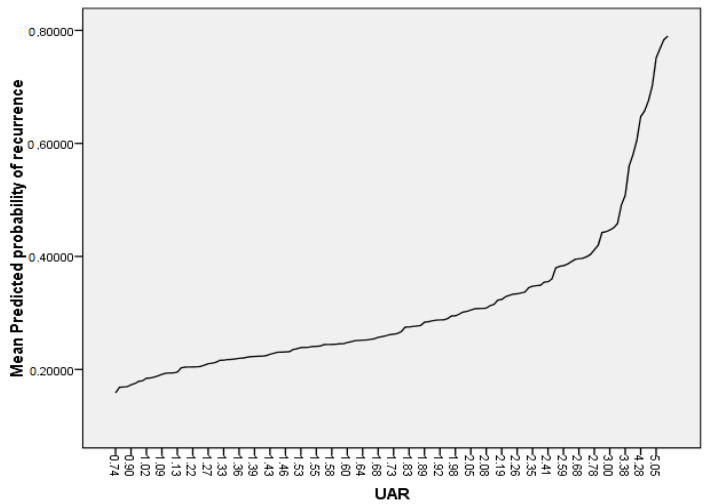
Relationship between probability of atrial fibrillation recurrence and uric acid/albumin ratio (UAR).

**Table 1 medicina-58-01872-t001:** Comparison of the demographic and clinical parameters between study groups.

	Total n = 170	Recurrence (+) n = 53	Recurrence (−) n = 117	*p* Value
Age, year	59.1 ± 11.7	61.3 ± 11.5	57.6 ± 11.4	0.07
Male, n (%)	96 (56)	29 (55)	67 (57)	0.44
Hypertension, n (%)	93 (54.7)	43 (81.1)	50 (42.7)	<0.01
Coronary artery disease, n (%)	41 (24.1)	26 (49.1)	15 (12.8)	<0.01
Heart Failure, n (%)	29 (17.1)	24 (45.3)	5 (4.3)	<0.01
COPD, n (%)	15 (8.8)	6 (11.3)	9 (7)	0.10
Diabetes mellitus, n (%)	24 (14.1)	12 (22.6)	12 (10.3)	0.03
Chronic kidney disease, n (%)	10 (5.9)	3 (5.6)	7 (6)	0.50
Smoking, n (%)	54 (31.8)	16 (30.2)	38 (32.5)	0.76
CHA_2_DS_2_-VASC score	1.8 ± 0.9	2.9 ± 1	1.4 ± 0.9	<0.01
AF duration, month	17 {30}	36 {43}	12 {22.5}	<0.01
AF pattern (PAF), n (%)	121 (71.2)	22 (41.5)	99 (84.6)	<0.01
Hemoglobin, g/L	14.2 ± 5.6	14.7 ± 9.7	13.9 ± 1.8	0.40
Serum creatinine, mg/dL	0.9 ± 0.2	0.9 ± 0.2	0.8 ± 0.2	0.17
LDL, mg/dL	116 ± 34.5	116.2 ± 31.1	115.8 ± 36.1	0.94
TSH, mIU/L	2.1 ± 1	2.3 ± 1	1.8 ± 0,9	0.25
CRP, mg/L	0.2 {0.25}	0.2 {0.3}	0.2 {0.2}	0.81
Uric acid, mg/dL	7.1 ± 2.3	8.1 ± 2.6	6.5 ± 2.1	<0.01
Albumin, g/dL	3.6 ± 0.6	3.5 ± 0.6	3.7 ± 0.5	0.03
UAR	2.03 ± 0.9	2.4 ± 0.9	1.8 ± 0.7	<0.01
LVEF,%	57.6 ± 13.7	48.4 ± 11	61.7 ± 8.9	<0.01
LA diameter	39.7 ± 6.3	44.1 ± 5.5	37.7 ± 5.6	<0.01
Left ventricular hypertrophy, n (%)	18 (10.6)	7 (13,2)	11 (9.4)	0.45
Mild aortic or mitral valve disease, n (%)	68 (40)	22 (42)	46 (39)	0.80
Duration of procedure, min	123.9 ± 22.5	121.8 ± 26.7	124.8 ± 20.3	0.41
Peri-ablation therapy				
Beta blocker/Ca channel blocker, n (%)	153 (90)	47 (89)	106 (90)	0.86
Amiodarone, n (%)	47 (27)	14 (26)	33 (28)	0.74
Propafenon, n (%)	74 (44)	22 (41)	52 (44)	0.69

Abbreviations: COPD, chronic obstructive pulmonary disease; AF, atrial fibrillation; LDL, low density lipoprotein; TSH, thyroid stimulating hormone; UAR, uric acid albumin ratio; LVEF, left ventricle ejection fraction; LA, left atrial.

**Table 2 medicina-58-01872-t002:** Multivariable regression analyses of the parameters which were found to be correlated with recurrence after univariable regression analyses.

	Unadjusted HR	95% CI	*p* Value	Adjusted HR	95% CI	*p* Value
LA diameter	1.15	1.08–1.21	<0.01	1.08	1.01–1.16	0.01
UAR	1.59	1.25–2.02	<0.01	1.36	1.06–1.75	0.01
CHA_2_DS_2_-VASC score	1.34	1.13–1.59	<0.01	1.02	0.83–1.25	0.81
AF pattern	0.29	0.16–0.51	<0.01	0.55	0.27–1.1	0.09
AF duration	1.01	1.01–1.02	<0.01	1.02	0.99–1.02	0.19

Abbreviations: LA, left atrial; UAR, uric acid albumin ratio; AF, atrial fibrillation.

## Data Availability

Not applicable.

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
