# Peer review of "Association of Uric Acid Albumin Ratio with Recurrence of Atrial Fibrillation after Cryoballoon Catheter Ablation"

_medicina, 2022, doi:10.3390/medicina58121872_

Round 1

Reviewer 1 Report

I applaud the authors for this interesting paper describing association of uric acid to albumin ratio and risk of recurrence of atrial fibrillation in patients following catheter ablation. It is a very interesting idea with a great scientific and clinical potential. 

The paper reads well however there are a number of grammatical and typographical errors throughout which need addressed.

I have made a number of points below which also should be addressed prior to publication in no particular order:

1. What was the baseline uric acid and albumin, levels. Can authors expand on correlation between these two.

2. The definition of paroxysmal and permanent atrial fibrillation seems unclear- can authors explain the arbitrary "termination by electrical or pharmaceutical cardio version" within 2 days and beyond 2 days of onset. As many patients with AF are asymptomatic and AF is frequently diagnosed incidentally, can authors expand how the day of onset was established and justify the "2 days" cut off period to distinguish between the two types.

3. Did all patients undergo Holter monitoring post ablation? (to detect asymptomatic occurrence), if not- can the authors provide details how many patients with recurrence were diagnosed based on symptomatic vs asymptomatic AF.

4. What was the baseline and post ablation medical therapy? How many patients were on antiarrhythmic medications, what were the types/classes and doses. Was there a difference between the two groups?

5. What was baseline BMI of patients, was there a difference between the two groups and was baseline BMI included in the prediction model?

6. The median follow up was 22 months- what was the min and max.

7. How many patients had valvular disease which could have increase their risk of AF?

8. AF pattern and AF duration were correlated with risk of recurrence in unadjusted analysis, but not adjusted. Can the authors propose a justification why this may be the case?

9. Figure 4 shows nearly exponential increase in risk of AF recurrence observed with UAR>3.0. It is not entirely clear whether this is due to display and x axis scale, or whether this is a true effect.

10. Can authors provide details of pattern of recurrence? E.g. were patients with paroxysmal AF more or less likely to have a recurrence of paroxysmal or permanent AF. Was there a difference between the two groups.

This study provides important information and highlights a potential use of UAR as a simple marker to identify patients with low or high chance of successful ablation. 

Author Response

Dear Editor,

We would like to thank the reviewers for their comments which helped us to improve the scientific value of the manuscript. The changes made are listed in the following section.

First reviewer

Q1. What was the baseline uric acid and albumin, levels? Can authors expand on correlation between these two.

A1. We provided the baseline uric acid and albumin levels. Also, we performed correlation analyses between two parameters as suggested. We added these findings in the results section.

Q2. The definition of paroxysmal and permanent atrial fibrillation seems unclear- can authors explain the arbitrary "termination by electrical or pharmaceutical cardio version" within 2 days and beyond 2 days of onset. As many patients with AF are asymptomatic and AF is frequently diagnosed incidentally, can authors expand how the day of onset was established and justify the "2 days" cut off period to distinguish between the two types.

A2. We corrected the definition of the paroxysmal and persistent atrial fibrillation and revised the methods section.

Q3. Did all patients undergo Holter monitoring post ablation? (to detect asymptomatic occurrence), if not- can the authors provide details how many patients with recurrence were diagnosed based on symptomatic vs asymptomatic AF.

A3. We performed routinely 24-h rhythm monitoring to all patients after ablation procedure. We added this point in the methods section.

Q4. What was the baseline and post ablation medical therapy? How many patients were on antiarrhythmic medications, what were the types/classes and doses. Was there a difference between the two groups?

A4. Antiarrhythmic drugs (amiodarone and propafenone) used before ablation was continued after ablation in the first 3 months. After 3 months, all antiarrhythmic drugs were stopped. We have mentioned this issue in the methods section. So as suggested, we have added peri-procedural rate control and antiarrhythmic therapy in the Table 1. There was no difference between groups and this comparison is included in the results section.

Q5. What was baseline BMI of patients, was there a difference between the two groups and was baseline BMI included in the prediction model?

A5. We don’t have BMI data because of the retrospective nature of the study. This information was added to the limitations of the study.

Q6. The median follow-up was 22 months- what was the min and max.

A6. Min 7 months and max 96 months. We added this finding in the results section.

Q7. How many patients had valvular disease which could have increase their risk of AF?

A7. Moderate and severe valvular heart disease were excluded. 68 (40%) patients had mild mitral or aortic valve disease and this parameter is included in the Table 1 as recommended.

Q8. AF pattern and AF duration were correlated with risk of recurrence in unadjusted analysis, but not adjusted. Can the authors propose a justification why this may be the case?

A8. After adjustment, both parameters’ p values are close to significance (0.05) and our study population is relatively small. Because of this, we did not find these two parameters correlated with recurrence after adjustment.

Q9. Figure 4 shows nearly exponential increase in risk of AF recurrence observed with UAR>3.0. It is not entirely clear whether this is due to display and x axis scale, or whether this is a true effect.

A9. It is a true graphical demonstration. It is found that mean predicted probability of recurrence increased apparently in patients with UAR higher than 3.0.

Q10. Can authors provide details of pattern of recurrence? E.g. were patients with paroxysmal AF more or less likely to have a recurrence of paroxysmal or permanent AF. Was there a difference between the two groups.

A10. It is so difficult to establish the pattern of the recurrence because some recurrences are asymptomatic and detected in the 24-hour rhythm monitoring. We cannot differentiate whether they are paroxysmal or persistent in nature. This information was added to the limitations of the study.

Reviewer 2 Report

Dear authors,

I have read with interest your article, and i have to congratulate the authors on quite well written and sound article. I was surprised that the UAR have had a higher predicitve valu ethan the left atrial diameter. This is something what i would emphasize both in the discussion and conclusion sections.

Of note, cryoballon technique is still in clinical use and very often put as a comparator to pulsed wave ablation.  

There are several issues to be solved., namely: 

1)    Introduction: line 34, this is actually not true for pulsed wave ablation (PFA), it should be corrected. More to the point, pulsed wave ablation is a balloon technique in a way more similar to cryoablation than radiofrequency ablation

2)    Line 67, ….which sustained over than 7 days…. Correct spelling (add „more“)

3)    Line 69-70 the sentece does not make any sense, should be completely rewritten

4)    Line 72 – rhythm monitor or ECG holter, not rhythm holter

5)    Line 124 thyroid not troid

Author Response

Dear Editor,

We would like to thank the reviewers for their comments which helped us to improve the scientific value of the manuscript. The changes made are listed in the following section.

Reviewer 2:

A1.  Introduction: line 34, this is actually not true for pulsed wave ablation (PFA), it should be corrected. More to the point, pulsed wave ablation is a balloon technique in a way more similar to cryoablation than radiofrequency ablation

Q1. We have removed the sentence in line 34.

Q2.  Line 67, …. which sustained over than 7 days…. Correct spelling (add „ more“)

A2 We have corrected the spelling

Q3. Line 69-70 the sentence does not make any sense, should be completely rewritten

A3: We have corrected the sentence

Q4. Line 72 – rhythm monitor or ECG holter, not rhythm holter

A4. We have corrected as 24-hour rhythm monitoring and revised the methods section as suggested

Q5 Line 124 thyroid not troid

A5: We have corrected the word asthyroid’

Best regards.

Round 2

Reviewer 1 Report

Thank you for providing the answers to the points and suggestions. 

This is very interesting article which raises important potential prognostic factor.